# Extending Video Masked Autoencoders to 128 frames

**Nitesh B. Gundavarapu**[1*]    **Luke Friedman**[1*]    **Raghav Goyal**[2*†]    **Chaitra Hegde**[1*]
**Eirikur Agustsson**[1]    **Sagar Waghmare**[1]    **Mikhail Sirotenko**[1]    **Ming-Hsuan Yang**[1]
**Tobias Weyand**[1]    **Boqing Gong**[1]    **Leonid Sigal**[2‡]
[1]Google Research    [2]University of British Columbia
{ngundavarapu,lbfried,cvhegde,eirikur}@google.com
{rgoyal14,lsigal}@cs.ubc.ca

## Abstract

Video understanding has witnessed significant progress with recent video foundation models demonstrating strong performance owing to self-supervised pre-training objectives; Masked Autoencoders (MAE) being the design of choice. Nevertheless, the majority of prior works that leverage MAE pre-training have focused on relatively short video representations (16 / 32 frames in length) largely due to hardware memory and compute limitations that scale poorly with video length due to the dense memory-intensive self-attention decoding. One natural strategy to address these challenges is to subsample tokens to reconstruct during decoding (or *decoder masking*). In this work, we propose an effective strategy for prioritizing tokens which allows training on longer video sequences (128 frames) and gets better performance than, more typical, random and uniform masking strategies. The core of our approach is an adaptive decoder masking strategy that prioritizes the most important tokens and uses quantized tokens as reconstruction objectives. Our adaptive strategy leverages a powerful MAGVIT-based tokenizer that jointly learns the tokens and their priority. We validate our design choices through exhaustive ablations and observe improved performance of the resulting long-video (128 frames) encoders over short-video (32 frames) counterparts. With our long-video masked autoencoder (LVMAE) strategy, we surpass state-of-the-art on Diving48 by 3.9 points and EPIC-Kitchens-100 verb classification by 2.5 points while relying on a simple core architecture and video-only pre-training (unlike some of the prior works that require millions of labeled video-text pairs or specialized encoders).

## 1    Introduction

Long video understanding has witnessed growing interest with various aspects of the problem having been explored by recent works [1, 2]. This includes incorporating (i) efficient attention mechanisms [3, 4], (ii) designing memory modules [5–7] to reason over context from the past, and (iii) approaching the problem from video-language perspective by proposing benchmarks [2, 8] and architectural choices that effectively leverage the interplay between the modalities [9–11, 8, 12].

The majority of recent models rely on foundational models pre-trained in a self-supervised manner based on videos and/or, for vision-language counterparts, video-text pairs; these foundational models, once pre-trained, can then be fine-tuned for a specific task at hand (*e.g.*, action classification). Masked

---

[*]Equal primary contribution
[†]Work done as Student Researcher at Google Research
[‡]Work done as Visiting Research Faculty at Google Research
[1]Authors now at Google DeepMind

38th Conference on Neural Information Processing Systems (NeurIPS 2024).

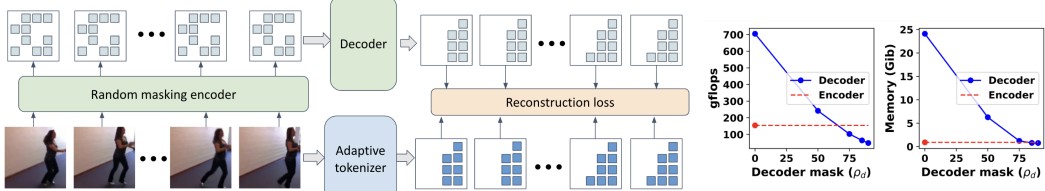

Figure 1: **Left: Proposed Long Video MAE Decoder Masking.** We leverage a jointly trained adaptive tokenizer and importance module to define a decoder mask and token targets for a video MAE pre-training strategy. The resulting sparsification in tokens (only 15%) allows pre-training with long videos (128-frames) and results in substantial performance gains. **Right: Decoder masking and memory in long-video (128 frames) pre-training.** We report memory and FLOPs per-device for a batch size of 1 using different decoder mask ratios and ViT-B architecture.

Autoencoders (MAE) have emerged as a simple and effective strategy to masked video modeling [13, 14] in this context. The standard MAE setup involves encoding a small fraction of visible / unmasked tokens (*e.g.*, 10%) of an input video, and decoding the remaining masked tokens (*e.g.*, 90%) using dense space-time attention decoder blocks. However, for longer video sequences, reconstructing all the masked tokens at the decoding stage quickly leads to out-of-memory (OOM) due to quadratic complexity of Transformers (feasible only for $< 64$ frames for moderate video resolution and current consumer hardware).

As a result, most existing MAE-based video understanding approaches focus on learning representations that encode only a few frames (16 [15] / 32 [14]) at a time. This limits their ability to understand actions and events spanning longer time horizons. Existing work has worked around the short context limitation by segmenting longer input video into short chunks and feeding them into the video model sequentially. Inference results from these chunks are then combined using late fusion. For example, "multi-crop" evaluations simply average-pool predictions over all chunks. However, this is clearly limiting. Long-context video models have the potential to understand longer actions (like complex diving routines), whole activities consisting of a sequence of actions and eventually entire story arcs.

Attempts to reduce computational cost and memory consumption, which bottleneck video MAE scalability, have recently started to emerge. These efforts can be characterized by either (i) reconstructing a subset of tokens at the decoding stage or *decoder masking* (VideoMAEv2 [14]), or by (ii) sub-sampling and focusing on a subset of "informative" tokens instead of using the entire set of spatio-temporal tokens (*e.g.*, focusing on measures of objectiveness [16] or motion [17] as proxies). The resulting gains have been shown to reduce training time and hardware requirement [17], and can be reinvested to scale model size [14]. However, scaling the input video along the temporal dimension has received little-to-no attention besides scaling input frames to a higher spatial resolution [14].

In this work, we build on [14] which performs masking at both the encoding and decoding stages. In particular, we focus on decoder masking considering its impact on memory and scalability, and propose a **content-dependent adaptive masking strategy** for videos in the MAE setup (see Figure 1). The core idea is a token-importance (or *saliency*) scheme, learned jointly with adaptive quantization, that establishes a rank-order of video tokens, which is then used to select higher-ranking tokens for decoding. The jointly learned quantization is shown to also be effective, compared to raw pixels, in defining the targets for reconstruction – further improving performance.

Our primary contributions are as follows:

1. We design a content-dependent adaptive masking strategy and demonstrate that given a low-token budget, it outperforms prior uniform and motion-based masking strategies.

2. The memory savings obtained from the low-token budget enables pre-training over long-videos (128 frames), allowing us to ask the question of exactly how much benefit long videos bring in the context of MAE pretraining. We observe that the long-video MAE (128 frames) model consistently outperforms short-video MAE (32 frames), as measured using downstream fine-tuning performance; including when a short-video MAE is fine-tuned with longer context (*e.g.*, 32 frame pre-trained MAE fine-tuned on 128 frames).

3. Leveraging these findings, we demonstrate state-of-the-art performance with our long-video MAE (LVMAE) approach on conventional action classification benchmarks that are known to require longer-range motion understanding (EPIC-Kitchens-100 [18] and Diving48 [19])

using standard ViT encoders and just one temporal crop – without relying on any language supervision or labels during pre-training.

## 2   Related work

**Masked Autoencoders (MAE).** Masked autoencoders have initially been proposed as an effective mechanism for learning self-supervised representations in masked image modeling (MIM) [20]. Following their success in the image domain, several video MAE-based representation learning extensions have been proposed. Specifically, ST-MAE [13] extends MIM to spatio-temporal patches in clips – treating space-time blocks as tokens, while VideoMAE [15] explores frame-level and tube-level masking strategies; both leverage an extremely high masking ratio ($\sim 90\%$) compared to image counterparts ($\sim 60\%$). Models that leverage joint image-video pre-training have also been introduced (*e.g.*, BEVT [21], OmniMAE [22]). However, scalability to high-resolution and long videos remains a substantial challenge, owing to the quadratic complexity of attention mechanisms.

**MAE Reconstruction Objectives.** While most video MAE models are trained to reconstruct the pixels of masked video patches [13, 15], there are a few notable exceptions. Mainly, BEVT [21] proposed to reconstruct discrete token targets obtained using a VQ-VAE [23] patch tokenizer, while MVD [24] focuses on masked feature modeling and distillation with high-level features as targets, obtained using a *teacher* model. Compared to BEVT, we see even greater gains when reconstructing discrete token targets by using a more powerful tokenizer (MAGVIT [25]) and an improved, jointly trained, tokenization and masking scheme.

**Efficiency by Token Reduction.** Sub-sampling salient tokens, based on a learned or an off-the-shelf strategy during training, is shown to give faster convergence and require less resources [17, 16, 26]. ObjectViViT [16] extracts object bounding boxes using an off-the-shelf object detector, and by placing emphasis on tokens belonging to objects was able to achieve lower token utilization and obtain competitive performance on the Epic Kitchens [18] and SSv2 benchmarks [27]. Token Learner [26] observes and corrects for token redundancy in higher layers of ViT [28] using learned modules.

Similar to the above approaches, we use token importance to inform our proposed masking. Differently, we don't sub-sample tokens, but selectively decode them during the training phase in a more *generalizable* dual masking setup. Concurrent EVEREST [17] is the most comparable to our work. It attempts to learn what tokens in the spatio-temporal volume are more informative using a heuristic based on distance in feature space (as a proxy for motion). In the short 16-frame regime they find that sub-sampling such tokens in MAE setup leads to lower memory and resource consumption, while maintaining performance. Different from [17], our token saliency scheme is learnt separately and independently in a MAGVIT tokenizer [25] setup, while EVEREST learns token-saliency during MAE pre-training itself. The latter approach can bias the method towards selecting *easy* tokens for the MAE reconstruction objective, undermining the learning, which is not an issue for our method. By learning token importance jointly with the tokenizer we incur a negligible extra cost, on top of the tokenizer computation, and as noted earlier get significant additional performance improvements by using the tokens as targets as opposed to RGB in [17].

**Motion in Video MAE**. Building on the above, attempts to identify and attribute relevance to video tokens involved in motion compared to static background has seen growing interest [29, 30], since motion is fundamental to understanding videos. In particular, MGMAE [29] and MGM [30] show that an informed encoder masking based on motion-cues at patch-level makes the MAE reconstruction task account for motion, and found faster convergence and improved performance on motion-centric video datasets such as SSv2 [27]. Specifically, MGM [30] uses H.264 codec [31] to extract patch-level motion vectors, and mask out tokens involved in motion during encoding, and MGMAE [29] uses online optical flow extractor (RAFT [32]), to warp and guide encoder masking using flow. Unlike these approaches, we do not model motion explicitly, and any apparent motion information in our mask is a byproduct of our data-driven token importance learning scheme.

**Long Videos and Masking**. As discussed above, masking tokens during training reduces memory requirement and this fact is exploited in several video understanding works for scaling model size [33, 34]. However, expanding these techniques for long-videos has received limited attention. Recently, LongViViT [9] leveraged random masking during contrastive pre-training on video-language tasks, and observed best performance-memory trade-off. Specifically, in order to convert short-video encoder to long-videos, they fine-tune the last four layers of ViViT [35] on long videos (128 frames) along with 75% random masking. Similar to LongViViT [9], we utilize masking to pre-train over long

videos. However, we focus on masked autoencoders instead of contrastive learning for long-videos and pre-train *all the layers* of our model, instead of partial freezing, in order to capture dense space-time correspondence in long-videos. Further, the overall training procedure of LongViViT [9] is considerably more complex requiring first pre-training with short videos and then further pre-training with long ones. We on the other hand, not only have a simpler scheme that can directly pre-train a MAE model with long videos (128 frames), but show that this is critical for improved performance.

**Long Videos and Memory**. Memory-based approaches [5, 6] attempt to form a compressed representation of the past activations or *memory*, which is then incorporated into current window or time step, effectively prolonging the context length over which reasoning and predictions are formed. Orthogonal to these works, we expand the local context of clips during pre-training from 16 to 128 frames; this can be combined with a memory module to increase the global context window.

## 3    Approach

We first give the background on MAE and the dual masking, initially introduced in [14]. We then focus on describing our proposed adaptive importance masking strategy and discuss how it can be leveraged to train with up to a 128 frame context window.

### 3.1    Background: MAE and Dual Masking

An input video $\mathbf{V} \in \mathbb{R}^{3 \times F \times H \times W}$ is first partitioned into non-overlapping spatio-temporal patches, and tokenized using a patch embedding layer (typically a `3DConv`) to give tokens $\mathbf{T} = \{T_i\}_{i=1}^{N}$, where $T_i \in \mathbb{R}^d$ is an $i^{th}$ token with added positional encoding, $N$ is total number of tokens, $d$ is the hidden dimension, and $F$, $H$, $W$ are number of frames, height and width respectively of the input video. Using an encoder mask $\mathcal{M}_e \in \{0,1\}^N$, the unmasked / visible tokens are selected $\mathbf{T}^u = \{T_i\}_{i \in (1-\mathcal{M}_e)}$, with $N^e$ the total number of unmasked encoder tokens. A vanilla ViT encoder [28] is applied to the unmasked tokens to give encoded visible tokens $\mathbf{Z} = \Phi_{enc}(\mathbf{T}^u)$. In the usual case, we obtain input to the decoder $\mathbf{Z}^c$ by combining the encoded tokens $\mathbf{Z}$ with learnable masked tokens $\mathbf{M} = \{M_i\}_{i \in \mathcal{M}_e}$, where $M_i \in \mathbb{R}^d$ is `[MASK]` token embedding with positional embedding. The overall objective is to reconstruct the masked-out tokens using the unmasked encoded tokens. However, with Dual Masking [14], a decoder mask $\mathcal{M}_d \in \{0,1\}^N$ is used to select tokens to reconstruct $\mathbf{Z}^c = \mathbf{Z} \cup \{M_i\}_{i \in (1-\mathcal{M}_d)}$, with $N^d$ the total number of unmasked decoder tokens where $N^e << N^d$ and $N^e + N^d << N$. Then, the combined tokens are reconstructed using a vanilla ViT decoder $\hat{\mathbf{V}} = \Phi_{dec}(\mathbf{Z}^c)$. Finally, the Mean Squared Error (MSE) loss is computed between the original and reconstructed pixels,

$$\mathcal{L} = \frac{1}{|\mathcal{M}_e \cap \mathcal{M}_d|} \sum_{i \in \mathcal{M}_e \cap \mathcal{M}_d} |\hat{\mathbf{V}}_i - \mathbf{V}_i|^2. \tag{1}$$

Alternatively, if vector quantized tokenization is employed, the loss becomes,

$$\mathcal{L} = \frac{1}{|\mathcal{M}_e \cap \mathcal{M}_d|} \sum_{i \in \mathcal{M}_e \cap \mathcal{M}_d} |\hat{\mathbf{V}}_i - VQ(\mathbf{V})_i|^2, \tag{2}$$

where $VQ(\cdot)$ is a vector quantization mapping typically trained separately using VQ-VAE or variants. In the proposed approach $VQ(\cdot)$ takes the form of adaptive FSQ-MagViT described in Section 3.3. Further, we want to highlight that encoder and decoder masking strategies need not be the same.

### 3.2    Masking Strategies

The above dual masking formulation requires both encoder and decoder masking. A number of strategies have been explored, but largely fall into two categories: *content agnostic* and *informative*.

*Content agnostic* masking strategies leverage either a fixed or randomized scheme which is agnostic of the video content. Fixed strategies comprise of (i) grid-based *uniform* masking [15], which keeps every $k$-th row/column along spatial dimension; and (ii) *frame* masking [14] where all tokens from every $k$-th frame are kept in an attempt to reduce temporal redundancy. The choice between the two depends on the assumptions regarding relative importance of spatial vs. temporal information.

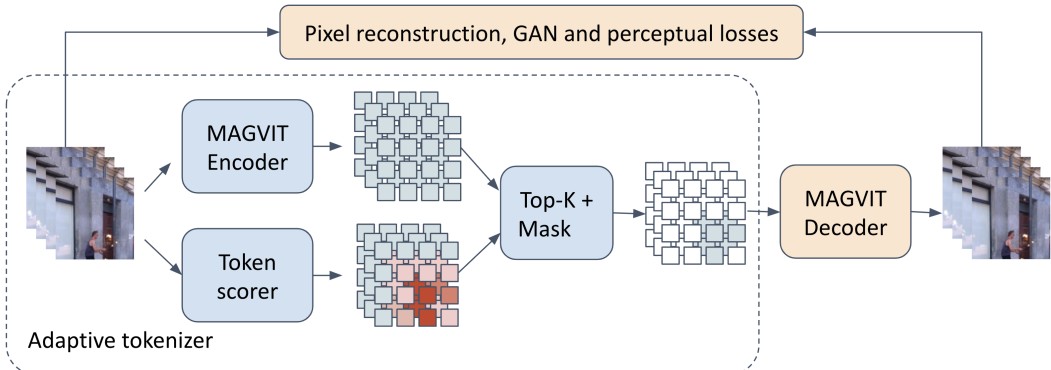

Figure 2: **Illustration of Adaptive FSQ-MagViT Training.** FSQ-MagViT adaptive tokenizer includes MAGVIT encoder and CNN-based token scorer with a differentiable top-$k$ selection layer designating importance of tokens. During tokenizer training unselected tokens zeroed out and video is reconstructed using MAGVIT decoder. We then freeze this adaptive tokenizer and use it to generate target tokens for scalable pre-training of video MAE.

(iii) Randomized strategies [13, 15] leverage sampling instead, which results in a form of data augmentation, since the same clip encountered in different epochs would result in a different mask. Under low-token budget, the above mentioned approaches are shown to perform competitively, and at the same time, gain memory efficiency [17].

Masking approaches mentioned above are effective, but at their core are sub-optimal as they fail to take into account the content of the video itself. Our approach falls into an *informative* masking class of strategies, which instead of sub-sampling overall set of tokens, leverage an importance function to produce a relative rank-order (or priority) on input tokens and use this ordering when selecting which tokens to encode/decode. We denote the token-importance (or saliency) as $\mathbf{S} \in [0, 1]^N$, where $N$ is the total number of tokens in an input video. In particular, $\mathbf{S}$ is a probability distribution over tokens, which gives us a rank-order of tokens. Letting $\mathbf{S}$ be a probability distribution allows both deterministic ($k$ most probable) and stochastic forms of informed masking. Prior work has relied on optical flow [29, 31] as proxy for importance, which in this work we leverage as one of our baselines. We, on the other hand, propose a more general adaptive FSQ-MagViT scheme that learns importance of tokens concurrently with tokenization (see Section 3.3), leveraging a MAGVIT [25] tokenizer.

While the above mentioned strategies can be used to generate both encoder and decoder masks in the dual masking MAE [14] (with the potential mild overhead of keeping the two sets of tokens disjoint), in this paper we focus on decoder masking in particular. This design choice is motivated by the observation that decoder masking has a much more significant impact on the memory usage and scalability of the overall MAE framework (see Figure 1) (right). Our focus on informative and effective decoder masking allows us to pre-train on long-video sequences (128 frames) – see Figure 1 (left). To this end we keep the encoder masking for most experiments in this paper relatively simple.

**Encoder masking**. Motivated by results in [15] for most experiments in the paper (unless otherwise stated) we use randomized *tube* masking with ratio of 90% (*i.e.*, encoder only sees 10% of patches).

**Decoder masking**. We generate `Adaptive` decoder mask $\mathcal{M}_d$ based on the intuition that reconstructing high-salient tokens is more meaningful for the MAE pre-training and for downstream fine-tuning tasks. We select top-$k$ salient tokens from $\mathbf{S}$ and set them as visible (indicated as 0's) in the decoder mask $\mathcal{M}_d \in \{0, 1\}^N$, where $k = (1 - \rho_d) \times N$ and $\rho_d$ is decoder masking ratio. In addition to selecting top-$k$ tokens, we also include a small fraction ($\rho_r$) of tokens sampled *randomly*, which we observed to improve performance, similar to ObjectViViT [16]. Therefore, $N^d = (1 - \rho_d) \times N + \rho_r \times N$. Illustration of resulted mask is shown in Appendix A.6.

## 3.3 Adaptive Finite Scalar Quantized VAE for Saliency and Reconstruction Targets

Our adaptive token selection module, shown in Figure 2, combines an effective CNN-based tokenizer and a differentiable top-$k$ selection layer.

**Tokenizer.** For the tokenizer we combine MAGVIT [25], which is a powerful 3D-CNN based video tokenizer and Finite Scalar Quantization (FSQ) [36], a simple alternative to VQ that easily scales

up to large codebooks. We refer to this combination as FSQ-MagViT and provide more details in Appendix A.5.1.

**Token Scorer** For the token selection, we learn a CNN-based Token scoring module followed by a differentiable top-$k$ layer, resulting in the mask which we apply to the FSQ-MagViT tokens. [1] More precisely, we embed the video to a feature space with two CNN layers, strided both spatially and temporally to match the resolution of the tokens from FSQ-MagViT encoder. From this CNN feature, we obtain pairwise Euclidean distances between spatial locations in adjacent frames, measuring how much each token in the $i$-th frame differs from the corresponding token in the $(i-1)$-th frame. This distance is considered as token importance as it signifies the extent of underlying change in the video as done in [17].

**Adaptive FSQ-MagViT** We get the soft-tokens from FSQ-MagViT and token importance from the token scoring module. We keep the tokens with the top-$k$ largest distances ($k = 768$) and mask out the remaining $N - k$ tokens, replacing their values with 0. For the special case of the first frame we keep all the tokens during training. These masked tokens are then fed through FSQ and the MagViT decoder. Similar to MAGVIT [25], we optimize for pixel reconstruction loss, GAN loss and perceptual loss by training the FSQ-MagViT and token scorer end-to-end.

### 3.4 Implementation details

First, we train our adaptive token selection module on Kinetics600 [37] data on 16 frame clips. Once trained, we keep this module frozen. Next, we follow standard MAE pre-training with a couple of changes (i) we reconstruct the top-$k$ tokens selected by the token selection module (and a small number of randomly selected tokens), and (ii) we reconstruct the latent embeddings from the tokenizer instead of RGB pixels. Note, we consciously choose to decouple the learning of token selection and MAE itself so that the gradients do not bias token selection towards selecting easily unmaskable tokens and vice-versa. Further implementation details are in Appendix A.5.1.

## 4 Experiments

We experiment with LVMAE on conventional video action classification benchmarks that have potential to benefit from long-video encoders: EPIC-Kitchens-100 (EK100) [18] and Diving48 (D48) [19]. EK100 [18] contains $\sim$ 90K video clips in total at 25 FPS with 3.7 secs avg. duration, and $\sim 15\%$ videos with at least 5 secs (=125 frames) duration. The task is to classify nouns (=300) and verbs (=97) together. D48 [19] contains videos with 158 avg. number of frames, and vary from 24 to 822 frames, and task to classify among 48 fine-grained dive categories [5]. Results on additional datasets are presented in Appendix A.2.

### 4.1 Decoder is the most memory intensive stage in long-video MAE

In Figure 1 (right), we show the memory and compute characteristics of ViT-B MAE architecture as we vary the decoder masking ratio for a video length of 128 frames. We fix the encoder masking ratio to 90% for this comparison. Even though the encoder has 3X more layers, decoder starts dominating for long videos due to the high number of tokens and quadratic scaling. We find that the decoder masking ratio should be close to the encoder masking ratio to respect compute and memory constraints imposed by the long-video pre-training regime.

### 4.2 Adaptive decoder masking strategy outperforms on short videos

In Table 1, we compare the proposed `Adaptive` masking with several alternative decoder masking strategies on short videos (32 frames) using both RGB pixels and FSQ-MagViT tokens as reconstruction targets on the EK100 dataset. We establish a baseline using the default MAE configuration which involves random tubes [14] as encoder mask (with mask ratio 90%), and decoding all the masked tokens, denoted as decoder mask ratio `None`. Then we experiment with the low-budget setting,

---

[1] We note that EVEREST[17] does non-differentiable top-$k$ selection on the low-level features of the MAE. In our case, we are learning the CNN top-$k$ module in parallel with FSQ-MagViT and need differentiable top-$k$ to be able to learn it end-to-end.

Table 1: **Decoder masking strategies on short-videos (32 frames) on EK100**. We report fine-tuning top-1 action classification performance of MAE pre-trained models using pixel (RGB) or token (FSQ-MagViT) reconstruction targets. For decoder masking, we consistently use $15\%$ as the token budget. We find that (1) compared to no decoder mask (`None`) with $100\%$ budget, uniform masking scheme (`Uniform` [14]), random masking scheme (`Random`), and decoder masking using Optical Flow (`Flow`) perform competitively with the lower token budget, and (2) we obtain best results with our proposed Adaptive decoder masking scheme (`Adaptive`). *Note that fine-tuning performance is reported on 1 temporal crop.*

| Saliency scheme | Masking | RGB | FSQ-MagViT |
|---|---|---|---|
| None | | 39.87 ±0.14 | 42.63 ±0.07 |
| Random | ✓ | 39.51 ±0.11 | 42.20 ±0.07 |
| Uniform [14] | ✓ | 39.92 ±0.14 | 41.73 ±0.05 |
| Flow | ✓ | 39.10 ±0.12 | 42.18 ±0.16 |
| EVEREST [17] | ✓ | 36.15 ±0.14 | 39.24 ±0.08 |
| Adaptive (Ours) | ✓ | **40.58** ±0.08 | **43.21** ±0.09 |

$N^d = 0.85N$ (i.e. $15\%$ token budget), where tokens to be decoded are selected using different saliency schemes described below.

**Random and Uniform**. The `Random` importance scheme selects tokens randomly from the spatio-temporal volume. For the `Uniform` saliency scheme, we form decoder-visible tokens by picking frames uniformly using a step size of 7 to select $1/7^{th}$ of frames or roughly $15\%$ token budget for decoding. We observe competitive performance to the baseline, consistent with VideoMAEv2 [14].

**Flow and EVEREST**. For the `Flow` importance scheme, we identify parts of the input video that contain motion. First, we obtain pixel-level Optical Flow $\mathbf{F} \in [-1,1]^{2 \times T \times H \times W}$ for the video using RAFT [32], containing both $x$ and $y$ displacement fields. We then use the pixel-level Flow to obtain a probability distribution of motion in token space to generate masks. Specifically, following the same patchify operation used for RGB video, for each spatio-temporal patch in Flow, we take the mean of absolute value for all pixels in the patch across $x$ and $y$ coordinates, and normalize across all patches to obtain $\mathbf{S} \in [0,1]^N$, where $N$ is the total number of tokens. For the proposed `Adaptive` scheme and the `Flow` scheme, we choose $\rho_d = 90\%$ and we add a small amount of random tokens, $\rho_r = 5\%$ as we found it improves the results. For EVEREST we follow details of [17].

Although, `Flow` and EVEREST strategies provide more informed token prioritization based on explicit motion vectors or learned pixel distance, respectfully, they don't perform as well as the proposed `Adaptive` masking approach. This is in part due to inaccuracies in prioritization that may result from optical flow when, for example, background motion is involved. Overall, we make a few observations: (1) with only $15\%$ decoder token budget, our proposed adaptive scheme bridges the gap with, and even improves on, vanilla VideoMAE which decodes all tokens with $100\%$ token budget (`None`), and (2) in addition to random and uniform schemes, that are content agnostic, our proposed strategy outperforms over content-informed approaches (`Flow` and EVEREST), demonstrating its effectiveness, and lastly (3) the above trends hold over **both** pixel and token reconstruction objectives. As a side note, our findings corroborate with other related works [25, 33, 24] that reconstructing higher-level targets outperform RGB pixel reconstruction for MAE. The masking strategy comparisons upon scaling to 128 frames are presented in Table 4d and illustrate similar trends.

### 4.3 Adaptive decoder masking enables pre-training over long videos

Using the best performing adaptive decoder masking strategy and FSQ-MagViT reconstruction targets, we reinvest the memory efficiency gained using just $15\%$ token budget towards extending the input number of frames to 128. Table 2 shows performance on long-videos (128 frames). We first highlight that default VideoMAE with no decoder masking encounters out of memory error due to large number of spatio-temporal tokens, demonstrating the need for decoder masking.

Overall, we observe that: (1) 128 frame pre-training outperforms 32 frames pre-training when fine-tuned on 128 frames; (2) 128 frame pre-training outperforms the typical 32 frames multi-crop evaluation. These two findings establish the significant benefit unlocked by our proposed model owing to long-video MAE pre-training, often ignored by previous models due to short context lengths.

Table 2: **Decoder masking enables training over long-video (128 frames)**. We report fine-tuning top-1 action classification performance of long-video MAE pre-trained models using token (FSQ-MagViT) as reconstruction target. *Note that 128 frames fine-tuning is performed using random-tube masking with* $20\%$ *masking and evaluation is reported on 1 temporal crop. Refer to Appendix A.5.3 for the fine-tuning details.*.

| Saliency scheme | 128 frames pre-training | | 32 frames pre-training | | | |
| --- | --- | --- | --- | --- | --- | --- |
| | Fine-tuning (128 frames) Eval (128 frames x 1 crop) | | Fine-tuning (128 frames) Eval (128 frames x 1 crop) | | Fine-tuning (32 frames) Eval (32 frames x 4 crops) | |
| | EK100 | D48 | EK100 | D48 | EK100 | D48 |
| None | out of memory | | 44.5 | 85.7 | 44.1 | 76.8 |
| Adaptive (Ours) | **47.3** | **87.9** | 45.0 | 83.2 | 45.0 | 75.7 |

## 4.4 Comparison with prior state-of-the-art works

In Tables 3a and 3b, we compare our proposed method, dubbed LVMAE, against the state-of-the-art on EPIC-Kitchens-100 and Diving48 respectively. For these comparisons, we first pre-train our model on unlabeled videos from Kinetics710 [38] data using 128 frames and adaptive masking strategy. This is followed by pre-training and fine-tuning on respective datasets at 128 frames similar to Sec.4.3. We additionally scale the encoder size to ViT-L. Further experiment details are in the Appendix. Note that existing SOTA methods use tailor-made architectures and/or pre-train their models using large-scale supervised pre-training data.

In Table 3a, we show that our proposed model improves current SOTA on EPIC-Kitchens Top-1 Verb classification by +2.5 points using standard ViT architecture and just a single crop. On the Top-1 Noun classification, our model lags behind MTV-B [39] pre-trained on 60 million labeled video clips and a specialized multi-view architecture, TAdaFormer [40] pre-trained on supervised Kinetics710 and recently published Avion [41] that pre-trains on large-scale egocentric data namely Ego4D. We would like to point out that large-scale labeled pre-training helps nouns more than verbs. The EPIC-Kitchens noun categories such as hands, gloves, spoon, knife etc. routinely appear as annotations in large-scale pre-training datasets such as ImageNet21k [42], Ego4D [43] etc. (e.g. 282/300 nouns from EPIC-Kitchens-100 also appear in ImageNet21k). As noted by Verbs-In-Action [44], verbs are relatively scarce in existing datasets. This explains why existing SOTA that use large-scale datasets excel at noun classification. On the other hand, our approach doesn't use large-scale pre-training datasets and learns long-range spatio-temporal dynamics to push SOTA on verb classification. Furthermore, in Table 3a, we find that if we add a supervised pre-training stage to our model using medium-scale dataset, we can bridge the gap on Noun classification while maintaining SOTA on Verb classification.

In Table 3b, we show that our proposed model improves the absolute state-of-the-art on Diving48 dataset which contains complicated diving sequences by 3.9 points. It's worthwhile to note that the current SOTA method, MC-ViT [5] effectively uses 27M video-text pairs while we use $\sim$ 1M unlabeled videos and just 15K labeled videos.

## 4.5 Ablation study

In Table 4a, we ablate our decoder masking strategy by pre-training models at 32 frames and report the performance on EPIC-Kitchens-100 dataset using a ViT-B backbone. Similar to [16], we find that reconstructing a small amount of random tokens helps in improving performance. Since reconstructing random tokens adds stochasticity to the sampling process, we hypothesize it can help with overfitting. On the other hand, it is sample inefficient to only reconstruct random tokens and combining these two approaches strikes a balance. We limit this ablation study to a sampling ratio of 15%, respecting memory constraints for extending to 128 frame pre-training.

In Table 4c, we gradually increase the number of frames and report the effect on EPIC-Kitchens-100 dataset. We find significant improvements in accuracy as we increase the number of frames from 16 to 32 to 64. However, as we move from 64 frames to 128 frames, the marginal improvement is small. This is expected as such videos form the tail-end of the distribution.

Table 3: **Comparison to State-of-the-art (SOTA).** In these tables we compare to a broad set of approaches, many of which use additional (labeled) data in pre-training or specialized modules.

(a) **Comparison to SOTA on EK-100.**

| Model | Extra Pre-training Data | Action | Verb | Noun |
|---|---|---|---|---|
| SlowFast [45] | K400 | 38.5 | 65.6 | 50.0 |
| IPL (I3D) [46] | K400 | 41.0 | 68.6 | 51.2 |
| ViViT-L/16x2 [35] | IN21K+K400 | 44.0 | 66.4 | 56.8 |
| MoViNet-A5 [47] | N/A | 44.5 | 69.1 | 55.1 |
| MeMViT-16, 16x4 [6] | K400 | 46.2 | 70.6 | 58.5 |
| MeMViT-24, 32x3 [6] | K600 | 48.4 | 71.4 | 60.3 |
| Omnivore (Swin-B) [48] | IN-(21K+1K)+K400+SUN | 49.9 | 69.5 | 61.7 |
| MTV-B [39] | IN21K | 46.7 | 67.8 | 60.5 |
| MTV-B$_{\uparrow 280^2}$ [39] | WTS-60M | 50.5 | 69.9 | 63.9 |
| TAdaFormer-B/16 [40] | K710 | 49.1 | 71.0 | 60.5 |
| TAdaFormer-L/16 [40] | K710 | 51.8 | 71.7 | 64.1 |
| *vision-language pre-training* | | | | |
| LaViLa (TSF-B) [49] | WIT + Ego4D | 46.9 | 69.0 | 58.4 |
| LaViLa (TSF-L) [49] | WIT + Ego4D | 51.0 | 72.0 | 62.9 |
| Avion (ViT-B) [41] | WIT + Ego4D | 49.1 | 70.0 | 59.8 |
| Avion (ViT-L) [41] | WIT + Ego4D | **54.4** | 73.0 | **65.4** |
| LVMAE (ViT-B) | None | 47.3 | 73.1 | 56.8 |
| LVMAE (ViT-B) | Unlabeled K710 | 47.0 | 73.0 | 56.3 |
| LVMAE (ViT-L) | Unlabeled K710 | 50.9 | **75.5** | 59.6 |
| LVMAE (ViT-L) | K710 | 52.1 | **75.0** | 61.8 |

(b) **Comparison to SOTA on Diving48.**

| Model | Pre-train | Top-1 |
|---|---|---|
| TimeSformer-L [50] | IN21K | 81.0 |
| VideoSwin-B [51] | IN21K | 81.9 |
| BEVT [21] | IN21K+K400 | 86.7 |
| SIFAR-B-14 [52] | IN21K | 87.3 |
| ORViT [53] | IN21K | 88.0 |
| AIM ViT-B/16 [54] | CLIP | 88.9 |
| AIM ViT-L/14 [54] | CLIP | 90.6 |
| MC-ViT-B [5] | ALIGN+LTIP+JFT +HT100M+VTP | 89.7 |
| MC-ViT-L [5] | ALIGN+LTIP+JFT +HT100M+VTP | 91.0 |
| Video-FocalNet-B [55] | K400 | 90.8 |
| LVMAE (ViT-B) | None | 87.8 |
| LVMAE (ViT-B) | Unlabeled K710 | 91.2 |
| LVMAE (ViT-L) | Unlabeled K710 | **94.9** |

In Table 4e, we compare our model's performance with the current SOTA model, Avion [41], on videos of different lengths using the EPIC-Kitchens-100 Verbs benchmark. For this study, we use the Large version of our model. We observe sustained performance improvements over SOTA with longer durations signifying our model's capability on longer sequences.

In Table 4b, we ablate the choice of targets for MAE pre-training. For this experiment, we first pre-train several models using 32 frames with no decoder masking. We find that when we shift from standard RGB targets to MAGVIT targets, the Noun-Verb Top-1 accuracy improves by 3.3%. Further, we notice a small drop from switching from MAGVIT targets to Adaptive FSQ-MagViT targets. In the last row, we show that when we use the adaptive decoder masking strategy with Adaptive FSQ-MagViT targets, we recover the performance. In effect, our proposed adaptive masking strategy

Table 4: **Ablation study**. We ablate a number of important design choices (please see text for details).

| MAE Pre-training (FSQ-MagViT) | Fine-Tuning |
|---|---|
| Sampling ratio Adaptive $(1 - \rho_d)$ + Random $(\rho_r)$ | Noun-Verb Accuracy |
| None | 42.6 |
| 15% + 0% | 42.5 |
| 10% + 5% | **43.2** |
| 0% + 15% | 42.2 |

(a) **Effect of masking ratios using adaptive scheme.**

| Masking | Target | Noun-Verb Accuracy |
|---|---|---|
| None | RGB | 39.9 |
| None | Standard MAGVIT | 43.2 |
| None | Adaptive FSQ-MagViT | 42.6 |
| Adaptive | Adaptive FSQ-MagViT | 43.2 |

(b) **Varying targets for MAE at 32 frames.**

| # of frames | Eval-Protocol | Noun-Verb Acc |
|---|---|---|
| 16 | 16 x 8 clips | 41.7 |
| 32 | 32 x 4 clips | 45.0 |
| 64 | 64 x 2 clips | 47.1 |
| 128 | 128 x 1 clips | **47.3** |

(c) **Number of frames ablation.**

| Saliency Scheme | EPIC-Kitchens | Diving48 |
|---|---|---|
| None | OOM | OOM |
| Random | 46.4 | 86.3 |
| Uniform | 45.6 | 85.2 |
| Flow | 46.3 | 86.3 |
| Adaptive (Ours) | **47.3** | **87.9** |

(d) **Masking strategy ablation at 128 frames with Adaptive FSQ-MagViT as targets.**

| Model | 0s-4s | 4s-8s | 8s-16s | 16s-32s | >32s |
|---|---|---|---|---|---|
| AVION [41] | 75.6 | 66.0 | 64.7 | 66.3 | 51.9 |
| LVMAE (Ours) | 77.8 | 67.0 | 66.2 | 72.2 | 57.7 |
| Relative Difference | +2.9% | +1.5% | +2.3% | +8.9% | +11.2% |

(e) **Comparison with SOTA on EPIC-Kitchens-100 Verbs at different video lengths.**

allows us to retain the performance boosts from MAGVIT targets at very high masking ratios, and thereby scale these gains to 128 frames effectively and surpass state-of-the-art. Unless otherwise mentioned, we always use Adaptive FSQ-MagViT as targets for all of our experiments.

In Table 4d, we compare our proposed adaptive decoder masking strategy with other strategies at 128 frames using FSQ-MagViT as targets. We find that our proposed adaptive masking is best as we scale the number of frames from 32 to 128 on both EPIC-Kitchens-100 and Diving48 datasets.

## 5 Limitations and broader impact

In this work, we restrict ourselves to relatively small datasets and model sizes and leave the exploration about large-scale pre-training, joint training of image and video datasets, higher-capacity models, etc. to future work. Furthermore, while the 128 frames in this work are a big leap from prior works' focus on 16 to 32 frames, we anticipate more significant improvements to handle longer videos. Using efficient decoders (and encoders) or combining our long local-context with memory offers an alternate path to scaling MAEs, which is orthogonal to our approach to some degree.

Improved long-video understanding can potentially revolutionize how users interact with video content since it can enable AI to reason across complex events and nuances. This potential will benefit accessibility, efficient content creation, recommendation, moderation, etc. Meanwhile, as with many machine learning models, our proposed method can be biased by the data. In addition, some applications (e.g., surveillance) might negatively impact society, and we urge users and researchers to deal with such use cases responsibly.

## 6 Conclusion

In this paper we present a relatively simple but highly effective adaptive masking strategy for video MAE pre-training that allows us to pre-train on long videos (128 frames). Our approach is based on a novel MAGVIT-based tokenization strategy that also learns importance of tokens. Further, we are the first, to our knowledge, to show that long-video MAE pre-training is not only possible but leads to better encodings. Using our approach we are able to achieve state-of-the-art downstream performance, despite using a simple architecture and video only pre-training.

# 7 Acknowledgements

We thank Chris Duvarney for finding better hyper-parameters to our models. We are grateful to Huisheng Wang, Nisarg Kothari, Philip Mansfield and Hartwig Adam for their continued support. We sincerely acknowledge Anurag Arnab and the Scenic team for the excellent framework.

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

# A Appendix

## A.1 Choice of tokenizer & quantizer

We choose MAGVIT [25, 56] (which has a 3D CNN encoder) because it is a strong video tokenizer architecture that has been shown to work both for generation and understanding [56] in prior works. For the choice of quantizer, prior works show that Lookup Free Quantizer (LFQ) [56] and Finite Scalar Quantizer (FSQ) [36] outperform traditional Vector Quantization (VQ) [23]. To study the effect of quantizer on our long-context MAE task, we first train several MAGVIT tokenizers with LFQ and FSQ quantizers at different codebook sizes, and then train MAE at 64 frames using our proposed decoder masking strategy. We present the corresponding results in Table. 5. We find that there is not much difference on the final EPIC-Kitchens-100 accuracy with these choices. However, FSQ with 18 bit codebook size shows the highest PSNR, the highest EK-100 accuracy and the second best FVD. Based on this result, we choose FSQ as our quantizer.

## A.2 Performance on additional datasets

In this section, we share supplementary results on two additional benchmarks, Something-Something-V2 [27] and FineGym288 [57].

### A.2.1 Something-Something-V2 benchmark

In Table 6, we test our proposed method on SomethingSomething-V2 [27] dataset, which has an average clip length of 3.8s at 12fps i.e. 46 frames on average, by training a Base sized model using our proposed decoder masking strategy with FSQ-MagViT as targets. We gradually increase the number of frames from 16 to 96 for this experiment.

The results match the trends we have seen on the other datasets in Table 4c and Table 2, with longer context providing a significant boost in final accuracy. There are diminishing returns from using more than 64 frames of context, as only a small number of examples from this dataset have clip length this long. Note that our performance exceeds both VideoMAE V1 and V2 [14, 15] for the base sized model, despite pre-training for fewer epochs (1600 vs. 2400) and using only a single crop at evaluation time.

### A.2.2 FineGym288 benchmark

FineGym288 [57] is a video classification benchmark that, similar to Diving48, tests the ability to categorize multi-second sports action sequences consisting of fine-grained motion, although in this case focusing on gymnastics. The original dataset has ~29K training examples and ~10K validation examples with video length ranging from 13 frames to 877 frames (average 47 frames). During dataset creation process, we add a 1s margin to the temporal action boundaries and increase the average video length to 107 frames, in order to exploit long-context.

In Table 7, we compare against SOTA when pre-training a 128 frame model with a ViT-L backbone and using additional unlabeled Kinetics-710 data (see section A.5.2). We outperform the current SOTA model, Temporal Cues Transformer [58], despite not using any additional temporal cues as in this work.

## A.3 Varying decoder budget

We ablate the effect of the decoder mask ratio on Diving48 top-1 accuracy while keeping the encoder mask ratio fixed at 90% in Table 8. For this experiment we use our adaptive masking strategy, a 128 frame context for pre-training and fine-tuning, and use adaptive FSQ-MagViT tokens as pre-training targets. For each experiment, we sample adaptive tokens and random tokens in 2:1 ratio. 15% token budget in the decoder yields the highest accuracy, significantly outperforming even the much more expensive setting with 50% token budget that requires >4x more memory than the 15% setting. This aligns with our observations in Table 1. *Note that for this experiment, we used improved hyper-parameters across all models, resulting in better results than presented in Table 4d.*

**Why does lower token budget outperform higher budget?** Videos contain redundant information, and we hypothesize that when we reconstruct a lower number of high importance (or higher rank

Table 5: **LFQ vs FSQ**. We compare tokenizers trained with different quantization schemes and report their reconstruction quality (PSNR, FVD) on Kinetics600 [37] benchmark and the corresponding MAE model's EPIC-Kitchens-100 top-1 accuracy.

| Quantizer | PSNR[K600] ↑ | FVD[K600] ↓ | Top-1[EPIC-Kitchens-100] ↑ |
|---|---|---|---|
| LFQ 14 bit codebook | 23.2 | 19.4 | 46.3 |
| LFQ 18 bit codebook | 22.6 | **14.7** | 46.2 |
| FSQ 14 bit codebook | 24.3 | 24.3 | 44.1 |
| FSQ 18 bit codebook | **25.1** | 19.1 | **46.4** |

Table 6: **Something-Something-V2 benchmark**. We report top-1 performance of our proposed MAE pre-training with decoder masking & FSQ-MagViT as targets while varying the number of frames.

| Number of frames | Decoder masking | Eval-Protocol | Top-1 Accuracy |
|---|---|---|---|
| 16 | None | 16 x 4 clips | 67.4 |
| 32 | None | 32 x 2 clips | 69.9 |
| 64 | 85% | 64 x 1 clips | **71.0** |
| 96 | 85% | 96 x 1 clips | 70.6 |

order) tokens during pre-training, the gradients can potentially be stronger than when we give equal importance to all tokens, and this would explain the stronger pre-trained encoder. However, we didn't notice this behavior consistently across datasets, namely on Something-Something-V2.

### A.4 Scaling model size vs number of frames

Given our decoder efficiency improvements, one can also scale model size instead of scaling on the number of frames axis. To study this effect, we first fix the memory budget and token budget to a reference Base sized model trained on 128 frames with a 15% token budget. Then we vary the model size, and for each model size, we maximize the number of frames that can fit in memory budget. In Table 9, we present top-1 accuracy, GFLOPs and the maximum frames on EPIC-Kitchens-100 dataset. We find that with larger model size and fixed memory budget, the accuracy improves despite lower frames but at the cost of significantly increased compute. We reiterate that all the three settings above are only made possible due to the memory savings from our proposed adaptive masking strategy. We leave further exploration in this direction to future work.

### A.5 Implementation details

#### A.5.1 Adaptive tokenizer

We train our adaptive token selection module following the training recipe from MAGVIT [25] on Kinetics600 dataset training on 16 frame clips, (model card in Table 10). For inference, we produce $8 \times 14 \times 14$ tokens along with the mask that determines the selected tokens.

Once trained, we keep this module frozen and use it across all our experiments. Note, we consciously choose to decouple the learning of token selection and MAE itself, so that the gradients do not bias token selection towards selecting easily unmaskable tokens and vice-versa. To obtain the adaptive mask for long-videos, we simply slide a window of 16 frames with a stride of 16 through the tokenizer and token scorer module, and concatenate the resulting tokens and importance masks.

#### A.5.2 Pre-training on long-videos (128 frames)

For our MAE pre-training we follow VideoMAE architecture [14] and employ a *vanilla* ViT-B encoder with full space-time attention and 12 layers and a 4 layer decoder with full space-time attention. This architecture will also produce $8 \times 14 \times 14$ tokens, matching the masks and latents produced by the adaptive tokenizer. Given a long video, we first compute the importance mask and select the $k = (1 - \rho_d) \times N$ tokens based on their importance from the video (plus $\rho_r \times N$ random tokens). This step can be computed offline and only needs to be run once per dataset. Next, we use a *tube* masking strategy and encode 10% tokens from the video along with their positional embeddings.

Table 7: **SOTA comparison on FineGym288**. We report results on the FineGym288 benchmark compared to current state-of-the-art methods.

| Method | Per-video Accuracy |
|---|---|
| TSM [59] | 83.1 |
| TQN [60] | 89.6 |
| VT-CE [61] | 90.1 |
| TCT [58] | 92.6 |
| LVMAE | **92.8** |

Table 8: **Varying decoder budget**. We report Diving48 top-1 performance and relative memory usage of our proposed MAE pre-training with decoder masking & FSQ-MagViT as targets.

| Decoder Budget | Top-1 Accuracy | Memory |
|---|---|---|
| 5% | 88.1 | 0.6x |
| 15% | 89.7 | 1x |
| 25% | 88.7 | 1.25x |
| 50% | 87.5 | 4.2x |

Then, we decode the chosen $N^d = k + \rho_r \times N$ tokens from these encoded tokens and learnable $MASK$ tokens along with respective positional embeddings. We make sure that no loss is applied on encoder-visible tokens. Unless otherwise mentioned, we initialize the MAE model from scratch and pre-train on respective downstream datasets for 1600 epochs. Full pre-training hyper-parameters are presented in Table 11.

For the case where we pretrain using additional K710 unlabeled data (see Tables 3a and 3b), we first pretrain an MAE model on K710 using our exact same recipe and hyperparameters as used elsewhere in this paper, except only for 800 epochs. We then pretrain a new MAE model on the downstream dataset (e.g. EPIC-Kitchens-100), initializing from the final checkpoint of this first model. We use identical hyperparameters when initializing from a pretrained checkpoint in this way as we do when training from scratch.

### A.5.3 Fine-tuning on long-videos (128 frames)

Finally, we fine-tune the pre-trained models on respective datasets using standard recipes detailed in Table 11 and report the performance metrics. In all cases we evaluate the final checkpoint of finetuning and average over runs with three different random seeds when reporting metrics to minimize variance. Fine-tuning batch size and epochs are typically much smaller than pre-training and hence we can afford to fine-tune our models with lower masking ratios (drop-token ratios) to fit in memory. In particular, we use a masking ratio of 20% in the encoder for 128 frame fine-tuning and 0% for 32 and 64 frame fine-tuning.

When fine-tuning the ViT-L backbone models on Diving48 and Epic-Kitchens-100, we make the following slight adjustments to hyperparameters:

1. We use a single layer of class attention [62] as the aggregation method when generating pre-logits as opposed to mean pooling. We found this slightly improved accuracy (+0.5 points on EPIC-Kitchens-100 Verbs and +1.2 points on Diving48).

2. We use 25% encoder masking instead of 20% encoder masking to avoid going OOM with the larger model.

3. For Diving48 we finetune for less steps (50 epochs instead of 200 epochs).

### A.5.4 Codebase and resources

We implement the code in Scenic [63] and run our pre-training experiments on 128 TPUv5e chips and fine-tuning experiments on 64 TPUv5e chips. Pre-training takes 24hrs and fine-tuning 16hrs in this setting for 128 frame models for EPIC-Kitchens-100.

Table 9: **Model size vs frames**. We report top-1 performance of our proposed MAE pre-training with decoder masking & FSQ-MagViT with different model sizes and maximum frames for that model size given a fixed memory budget.

| Model Size | GFLOPs (relative) | Max frames | EPIC-Kitchens-100 Top-1 |
|:---:|:---:|:---:|:---:|
| Small | 0.43x | 144 | 39.1 |
| Base | 1x | 128 | 47.3 |
| Large | 1.52x | 80 | 48.9 |

Table 10: Adaptive Tokenizer Model Card

| Config | Pre-training | Inference |
|---|:---:|:---:|
| Number of frames | 16 frames, frame stride 1 | |
| Spatial resolution | $128 \times 128$ | $112 \times 112$ |
| Model size[25] | B | B |
| Base channels[25] | 64 | 64 |
| VQVAE channel multipliers[25] | 1, 2, 2, 4 | |
| Discriminator channel multipliers[25] | 2, 4, 4, 4, 4 | |
| Latent spatiotemporal shape | $8 \times 16 \times 16$ | $8 \times 14 \times 14$ |
| Vocabulary size | $2^{18}$ using FSQ[36] with $[8, 8, 4, 4, 4, 4, 4]$ levels | |
| Embedding dimension | 8 | 8 |
| top-k for token selection | 768 | 15% |
| Batch size | 256 | 256 |
| Peak learning rate | $10^{-4}$ | - |
| Learning rate schedule | linear warm up and cosine decay | - |
| Optimizer | Adam with $\beta_1 = 0$ and $\beta_2 = 0.99$ | - |
| Generator loss type | Non-saturating | - |
| Generator adversarial loss weight | 0.1 | - |
| Perceptual loss weight | 0.1 | - |
| Discriminator gradient penalty | r1 with cost 10 | - |
| EMA model decay rate | 0.999 | - |

## A.6 Visualization of adaptive mask

In Figure 3 we visualize our adaptive tokenizer. We can see it focuses the tokens where the relevant motion is happening. In Figure 4, we show qualitative comparisons of our adaptive mask with other alternative masking strategies. As expected, optical-flow based masks do very poorly with pure camera motion as they cannot distinguish camera motion from foreground motion. In addition, we find that our mask places more relevance to foreground.

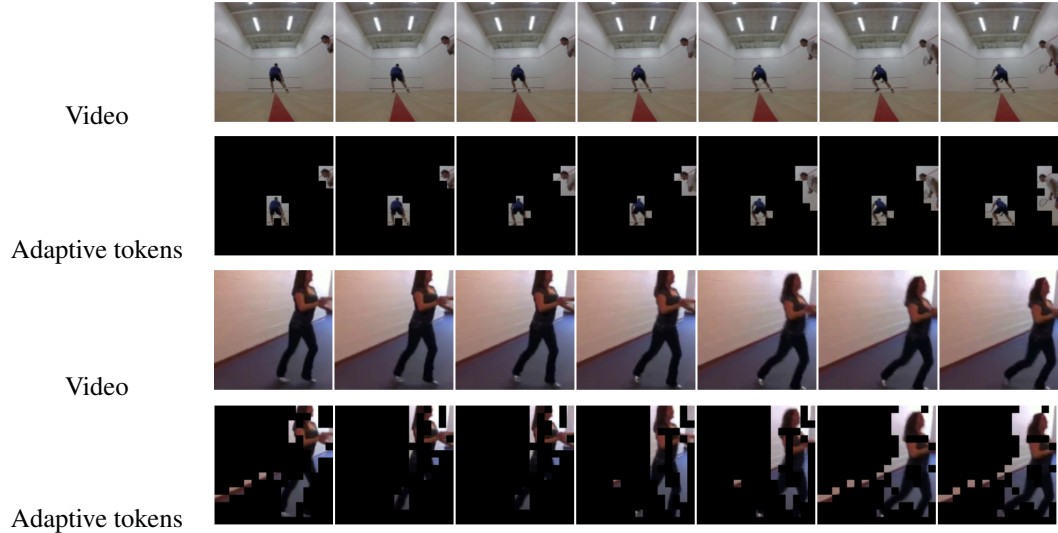

Figure 3: Our Adaptive tokenizer visualized. We visualize the tokens masks by masking the corresponding input video (repeating frames to match the latent temporal dimension).

Table 11: Model Card with detailed model architecture and training setups for Base size experiments on Diving48 and EPIC-Kitchens.

| Dataset | Pre-training | Fine-tuning |
|---|---|---|
| inputs | pixels | |
| targets | tokens | classes |
| encoder (layers, heads, MLP dim) | ViT-B $(12, 12, 3072)$ | |
| encoder input shape $(t{\times}w{\times}h{\times}c)$ | $(1 - \rho_e) * 32|128{\times}224{\times}224{\times}3$ | |
| tubelet dimensions $(w{\times}h{\times}t)$ | $16{\times}16{\times}2$ | |
| encoder output shape $(t{\times}w{\times}h{\times}c)$ | $(1 - \rho_e) * 16|64{\times}14{\times}14{\times}768$ | |
| decoder (layers, heads, MLP dim) | ViT $(4, 4, 1536)$ | None |
| decoder output shape $(n{\times}c)$ | $(1 - \rho_d + \rho_r)*16|64 * 14 * 14{\times}384$ | None |
| optimizer | Adam | Momentum |
| optimizer momentum | $\beta_1 = 0.9, \beta_2 = 0.95$ | $\beta = 0.9$ |
| weight decay | 0.05 | 0.0 |
| learning rate | $1.5e - 4$ | 0.5 |
| learning rate schedule | cosine decay | |
| warmup epochs | 40 | 2.5 |
| epochs | 1600 | EK: 50, Diving: 200 |
| augmentation | None | Jitter-Scale, Mixup, RandAug |
| batch size | 1024|512 | 64 |
| label smoothing | 0.1 | 0.2 |
| dropout | 0.1 | 0.0 |

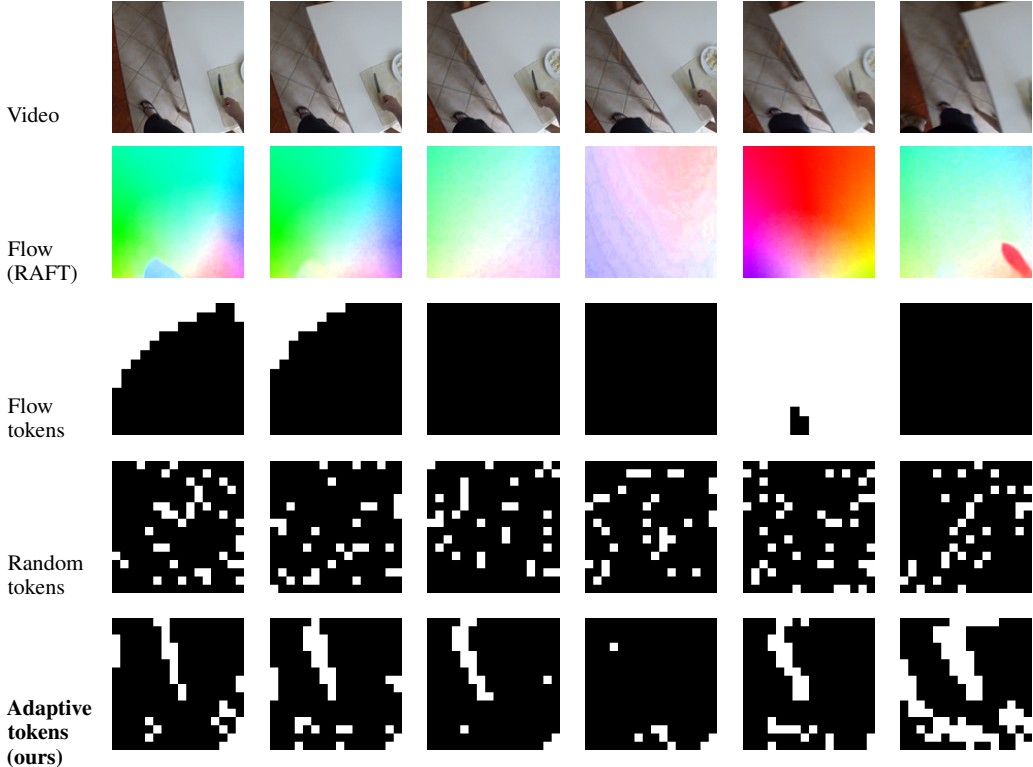

Figure 4: Token selection strategies visualized. We can see that flow based token selection can be dominated by large background motion. Randomly selected masks are unable to focus the tokens on the interesting parts of the video. In contrast, we see that the adaptively selected tokens reflect well what is changing in the video.

