# OpenReview forum: "Extending Video Masked Autoencoders to 128 frames"
_NeurIPS.cc/2024/Conference — NeurIPS 2024 poster_

### Official Review · Reviewer_EcJb · 2024-07-12

**Soundness:** 3
**Presentation:** 2
**Contribution:** 3
**Rating:** 5
**Confidence:** 4

**Summary:**

This paper studies the MAE pretraining of long videos. To address the challenges of hardware memory and compute limitations, this paper propose an effective strategy of decoder masking, subsampling tokens as reconstruction targets. This strategy leverage the MAGVIT-based tokenizer, prioritizes the most important tokens and uses quantized tokens as reconstruction objectives. The proposed decoder masking strategy outperforms previous masking strategies on EK-100 and Diving48. With 128 frames as input for pretraining and finetuning, the video encoder obtain better performance compared with the short-video (32 frames) pretraining.

**Strengths:**

1. Study the MAE-based pretraining of long videos. The proposed adaptive decoder masking strategy solves the OOM problem of long-video training and beats some previous masking strategy on downstream tasks.

2. Compared to 32 frames MAE-based pretraining, the proposed method with 128 frames obtains better performance on EK-100 and Diving48.

**Weaknesses:**

1. This paper only shows the results of 32 frames input and 128 frames input. To figure out the best setting of input frame number, it would be better to show experimental results when increasing the frame number from 16 to 128 gradually for pretraining and finetuning (while keeping the similar computational cost at inference).

2. This paper does not include any experiments or theory analysis to explain why to convert MAGVIT to FSQ-MAGVIT for reconstruction targets and adaptive token selection.

3. Only 2 downstream tasks are used for the main experiments.

**Questions:**

1. Why does this work convert MAGVIT to FSQ-MAGVIT for reconstruction targets and adaptive token selection?

2. Results on other long-form video downstream tasks.

**Limitations:**

Yes.

---

> ### Author Rebuttal · Authors · 2024-08-07
>
> We appreciate the reviewer’s valuable feedback on our work, and for noticing the effectiveness of our method in improving both pre-training efficiency and downstream accuracy. Below, we address some of the concerns and questions raised by the reviewer.
>
> ### Q1. Gradually increasing the number of frames:
> Our motivation is to study whether we can effectively pre-train MAE on long-videos and whether such pre-training helps downstream tasks. For this, we selected 128 frames as our target (as it is 8x the typical MAE pre-training frames already), and designed our method around achieving this target. Post-facto, we agree that it’s interesting to study the effect of varying number of frames. To satisfy reviewer’s curiosity, we present some related findings:
>
> In the table below, we ablate the effect of increasing the number of frames while keeping the mask ratio fixed at 15%. During inference, we ensure that the model sees the same number of frames and report Epic-Kitchens-100 action Top1 Accuracy.  We clearly observe a trend of the accuracy improvement with the number of frames similar to what we presented in Table 2 of the main paper.
>
> |Pre-training frames|Masking|Eval protocol|Epic-Kitchens-100|
> |-----|---|---|---|
> |16|15%|16 x 8 clips|41.7|
> |32|15%|32 x 4 clips|45.0|
> |64|15%|64 x 2 clips|47.1|
> |128|15%|128 x 1 clip|47.3|
>
>
> Below, we repeat the same experiment without decoder masking and observe OOM for 64 and 128 frames, whereas our proposed decoder masking method unlocks additional gains in the previous table. This further demonstrates the effectiveness of our method.
>
> |Pre-training frames|Masking|Eval protocol|Epic-Kitchens-100|
> |----|---|---|----|
> |16|None|16 x 8 clips|41.3|
> |32|None|32 x 4 clips|44.1|
> |64|None|64 x 2 clips|OOM|
> |128|None|128 x 1 clip|OOM|
>
>
> ### Q2. "Why does this work convert MAGVIT to FSQ-MAGVIT for reconstruction targets and adaptive token selection?"
> We thank the reviewer for raising this question. We have now added a justification in the [common response](https://openreview.net/forum?id=bFrNPlWchg&noteId=nljvnUQeki) on our choice of tokenizer and the quantizer, and we request the reviewer to consult the same. Our studies showed that our choice of FSQ achieved the highest PSNR and the highest downstream EK100 accuracy, however the differences in downstream EK100 accuracies are not much with different choice of quantizers.
>
>
> ### Q3. Only 2 downstream tasks are used for the main experiments.
>
> During the rebuttal period, we additionally tested our method on Something-Something-v2 dataset. We presented the corresponding results in the [common response](https://openreview.net/forum?id=bFrNPlWchg&noteId=nljvnUQeki) and observe the same trend as seen in Table 2 of the main paper.

---

> > ### Comment · Reviewer_EcJb · 2024-08-12
> >
> > Thanks for the authors' effort in the rebuttal. The response has addressed my first question (in the weaknesses). However, the authors have not provided experiments between different tokenizers under various ratios of adaptive masking, and the ablation studies include only a limited range of downstream tasks. I am still concerned about whether using FSQ in long-video MAE methods is necessary compared to a standard MAGVIT. Additionally, the experiments do not include various long-form video tasks, such as video action segmentation, temporal action localization, and spatiotemporal action detection. In summary, while the research topic is meaningful, I concerns that the experiments in this paper could not fully demonstrate the effectiveness and scalability of this method. Therefore, I would lower my rating slightly.

---

> > > ### Author Response · Authors · 2024-08-13
> > > **Additional clarifications on tokenizer choice**
> > >
> > > We thank the reviewer for such timely feedback. We agree with the reviewer that our exposition on the tokenizer can be improved further. However, we would like to point out that the focus of our work has been to find a way to effectively scale MAEs to 128 frames and we saw our tokenizer explorations as a means to achieve this goal. In addition, since tokenizer is the first stage of our pipeline, it is expensive to undertake ablations on tokenizer, especially, when we only see very limited gains when we switch tokenizers (as shared in common rebuttal). So, we chose the best tokenizer from literature [1] and picked one of the two best quantizers [1, 2]. Nevertheless, we are glad to see the reviewer's interest in our work and appreciate the experiment suggestions. We hope our response below addresses your concerns:
> > >
> > >
> > > `the authors have not provided experiments between different tokenizers under various ratios of adaptive masking`
> > >
> > > We thank the reviewer for this suggestion. We would first like to clarify that our tokenizer indeed sees different masking ratios during training as we fix the token budget at batch-level (on a batch size of 256) instead of at example level, and hence it can work at varying mask levels in the MAE training stage. For this reason, we treated the token budget of the tokenizer as a hyper-parameter and chose it to be 35% so that it can support a range around that value. We agree with the reviewer that both the choice of the tokenizer and the token budget can be further optimized to boost the performance of our method. We didn’t pursue this route as our focus was on enabling an effective strategy for long-video pre-training, and leave the exploration on tokenizers for understanding tasks to future works.
> > >
> > > `I am still concerned about whether using FSQ in long-video MAE methods is necessary compared to a standard MAGVIT`
> > >
> > > We chose FSQ because it's simple and performs well on reconstruction (see below for detailed results). However, we stress that the choice of quantizer is not critical for the downstream performance of our method, and we hence also do not make any claims encouraging FSQ in our work. It is likely that the crux of our paper and all the key findings may very well be unchanged had we picked LFQ.
> > >
> > > Below, we evaluate our pre-trained FSQ and LFQ tokenizers by varying the masking ratio to see how they respond. We expect the PSNR to increase (and FVD to decrease) as the token budget increases. We find that FSQ adheres to this expected behavior better than LFQ does. Especially at a lower token budget, performance of FSQ is better than LFQ. This criteria is well-suited for decoder masking where we use very high masking ratios.
> > >
> > >
> > > |Token budget|LFQ-PSNR | LFQ-FVD|FSQ-PSNR| FSQ-FVD|
> > > |-------------|-----|------|-----|-----|
> > > |15%|19.2|174.3|23.0|60.6|
> > > |20%|21.1|56.1|24.2|29.0|
> > > |25%|22.8|25.1|24.9|18.3|
> > > |30%|24.3|13.8|25.4|13.4|
> > > |35%|25.4|9.2|25.8|10.7|
> > > |40%|25.7|7.9|26.1|9.6|
> > > |45%|25.0|8.9|26.3|9.0|
> > > |50%|24.0|11.7|26.4|9.0|
> > >
> > >
> > > However, despite the reconstruction capacity, in our previous ablations we have shown that the MAE pre-training is not highly sensitive to the choice of quantizer in the Magvit.
> > > |Quantizer|EK-100|
> > > |----|---|
> > > |LFQ |46.2|
> > > |FSQ |46.4|
> > >
> > >
> > >
> > > [1] Yu, Lijun, et al. "Language Model Beats Diffusion--Tokenizer is Key to Visual Generation." arXiv preprint arXiv:2310.05737 (2023). \
> > > [2] Mentzer, Fabian, et al. "Finite scalar quantization: Vq-vae made simple." arXiv preprint arXiv:2309.15505 (2023).

---

> > > ### Author Response · Authors · 2024-08-13
> > > **Additional task types**
> > >
> > > We provided our results on short-clip (Something-Something-V2) and long-clip classification tasks (EPIC-Kitchens-100, Diving48) and demonstrated SoTA performance. In response to reviewer 9WAZ, we also provided a breakdown of our results according to duration of the videos, and found that [our method outperforms SoTA especially on the longer duration clips](https://openreview.net/forum?id=bFrNPlWchg&noteId=DvCiSrKkm7) i.e. >16s in length.
> > >
> > > One key issue that we faced when selecting datasets for evaluation is the lack of long-form video benchmarks. Many of the tasks that operate on long videos only require short video-contexts [1, 2, 3] and can be solved by sliding a short-clip window over the video lengths. For example, a single frame is shown to be enough to answer QA tasks in ActivityNet [2], which is also a widely used benchmark for temporal action localization. Recently, new benchmarks have come up that are shown to require a longer temporal certificate [1, 4, 5], but unfortunately they require multimodal understanding, which is beyond the scope of this paper, and we leave them to future works.
> > >
> > > [1] Mangalam, Karttikeya, Raiymbek Akshulakov, and Jitendra Malik. "Egoschema: A diagnostic benchmark for very long-form video language understanding." Advances in Neural Information Processing Systems 36 (2024). \
> > > [2] Lei, Jie, Tamara L. Berg, and Mohit Bansal. "Revealing single frame bias for video-and-language learning." arXiv preprint arXiv:2206.03428 (2022). \
> > > [3] Buch, Shyamal, et al. "Revisiting the" video" in video-language understanding." Proceedings of the IEEE/CVF conference on computer vision and pattern recognition. 2022. \
> > > [4] Wang, Weihan, et al. "LVBench: An Extreme Long Video Understanding Benchmark." arXiv preprint arXiv:2406.08035 (2024). \
> > > [5] Rawal, Ruchit, et al. "Cinepile: A long video question answering dataset and benchmark." arXiv preprint arXiv:2405.08813 (2024).

---

> > > > ### Author Response · Authors · 2024-08-14
> > > >
> > > > We hope that our response clarifies the rationale behind tokenizer choice and the choice of task types. We also hope that the break-down of our performance improvements along the dimension of video durations further substantiates our claims. We would like to request the reviewer to reevaluate our work in light of our response.

---

> ### Author Response · Authors · 2024-08-10
> **Please let us know whether we address all the issues**
>
> Dear reviewer,
>
> Thank you for the comments on our paper.
>
> We have submitted the response to your comments and a common response. Please let us know if you have additional questions so that we can address them during the discussion period. We hope that you can consider raising the score after we address all the issues.
>
> Thank you

---

### Official Review · Reviewer_UhsZ · 2024-07-13

**Soundness:** 4
**Presentation:** 4
**Contribution:** 3
**Rating:** 6
**Confidence:** 5

**Summary:**

The paper proposes a novel MAGVIT-based adaptive tokenizer & masking module to extend VideoMAE to 128 frames. The tokenizer & masking module is individually trained and applied offline, making it possible for VideoMAE to reconstruct sparser (but important) and more semantically informative targets. The experimental results show that the approach allows for significant memory savings, enabling pre-training on longer video clips and leading to improved performance on downstream long video understanding tasks.

**Strengths:**

1. The paper is well-written and easy to follow, with clear explanations of the proposed method and experimental results.
2. The paper addresses an important issue in video understanding, namely the scalability of video masked modeling for longer sequences. The approach in this paper is relatively orthogonal to prior work in the field.
3. The proposed module is individually trained and applied offline, which avoids the training difficulties and extra overhead in videomae pretraining that come with the complex designs.
4. The proposed module provides both decoder masking and high semantic information tokens, which saves computational overhead and improves model performance. The effectiveness of each part is verified through ablation experiments.

**Weaknesses:**

1. The major weaknesses of this paper include the limited dataset sizes, small model sizes, and narrow range of task types, as mentioned in Section 5. I'm interested in the model's performance on action detection and temporal action detection tasks.

**Questions:**

1. In Table 4a, the 10%+5% masking even exceeds the case without decoder masking, and perhaps the authors could provide more explanation for this phenomenon.

**Limitations:**

The author discusses the limitations of the work, which I guess are largely due to resource limitations.

---

> ### Author Rebuttal · Authors · 2024-08-07
>
> We appreciate the reviewers comments on our work being well-written, its importance in the field of video understanding, orthogonality to existing works, and that the effectiveness of our approach is verified through ablation experiments. Below, we address the additional questions raised by the reviewer.
>
> ### Q1. Limited dataset sizes and small model sizes
>
> We tested our approach on conventionally used model sizes that include Base (B) and Large (L) transformer variants on two of the widely used datasets, namely EPIC-Kitchens-100 and Diving48. Although there exists literature [1, 2, 3, 4] on large-scale pre-training and large model sizes for video understanding, we focussed our study on answering the question of how we can effectively pre-train MAEs on 128 frames and whether such pre-training helps. Extending our approach and adopting it in large-scale models is definitely an exciting prospect that we would like to explore, however it is beyond the reach of the current work.
>
> ### Q2. Limited and narrow range of tasks
>
> During the rebuttal period, we additionally tested our method on Something-Something-v2 dataset. We presented the corresponding results in the [common response](https://openreview.net/forum?id=bFrNPlWchg&noteId=nljvnUQeki) and observe the same trend as seen in Table 2 of the main paper. Due to time/resource constraints of the rebuttal period we were not able to run similar ablations for models with a VIT-L backbone and for different task types including temporal action localization, but we hope to include these experiments as well in a future version.
>
> ### Q3. “10%+5% masking even exceeds the case without decoder masking”
> We appreciate that the reviewer noticed this. Videos contain redundant information, and we hypothesize that when we reconstruct a lower number of high importance (or higher rank order) tokens during pre-training, the gradients can potentially be stronger than when we give equal importance to all tokens, and this would explain the stronger pre-trained encoder. However, we didn’t notice this behavior consistently across datasets, namely on SSv2. We will include this discussion in the paper.
>
>
> [1] Wang, Yi, et al. "Internvideo: General video foundation models via generative and discriminative learning." arXiv preprint arXiv:2212.03191 (2022). \
> [2] Li, Kunchang, et al. "Unmasked teacher: Towards training-efficient video foundation models." Proceedings of the IEEE/CVF International Conference on Computer Vision. 2023. \
> [3] Zhao, Long, et al. "Videoprism: A foundational visual encoder for video understanding." arXiv preprint arXiv:2402.13217 (2024). \
> [4] Wang, Yi, et al. "Internvideo2: Scaling video foundation models for multimodal video understanding." arXiv preprint arXiv:2403.15377 (2024).

---

> > ### Author Response · Authors · 2024-08-14
> > **Please let us know whether we address all the issues by the end of discussion period**
> >
> > Dear reviewer,
> >
> > Thank you for the comments on our paper.
> >
> > We have submitted a response to your comments and a common response. As the other reviewers have participated in the discussions, we would like to ask you to let us know whether you have additional questions.
> >
> > We hope that you can consider raising the score after we address all the issues.
> >
> > Thank you

---

> ### Author Response · Authors · 2024-08-10
> **Please let us know whether we address all the issues**
>
> Dear reviewer,
>
> Thank you for the comments on our paper.
>
> We have submitted the response to your comments and a common response. Please let us know if you have additional questions so that we can address them during the discussion period. We hope that you can consider raising the score after we address all the issues.
>
> Thank you

---

### Official Review · Reviewer_9WAZ · 2024-07-14

**Soundness:** 4
**Presentation:** 4
**Contribution:** 2
**Rating:** 3
**Confidence:** 4

**Summary:**

This video understanding paper extends the video mae idea to a longer 128 frames. They use the MAGVIT tokenizer to achieve this and test the approach on Diving-48 and epic kitchens. Both achieved an improved score despite using a pretty standard network and pre-training

**Strengths:**

The use of a MAGVIT encoder masking strategy within a masking strategy of videomae
The use of FSQ instead of VQ for the codebook encoding

**Weaknesses:**

The performance for EK-100 isn't that great.
The approach needs to be tested on longer sequence videos, maybe to make a more significant difference than the current two.
There is limited innovation on the actual network used

**Questions:**

Why does the approach only make sota for the verbs?
What is the limit on the length of tokens

**Limitations:**

The idea of focusing on more extended encodings is interesting, but the dataset performance doesn't seem to make it worth it.

---

> ### Author Rebuttal · Authors · 2024-08-07
>
> ### Differentiating EK-Noun SoTA with EK-Verb SoTA
> We appreciate the reviewer’s feedback and agree that our EK-100 noun accuracy is below SoTA. However, we would like to humbly point out that SoTA methods for EK-Noun rely on large-scale pretraining, while ours does not. The EK-100 noun categories such as hands, gloves, spoon, knife etc. routinely appear as annotations in large-scale pre-training datasets such as ImageNet21k [1], Ego4D [2] etc. (282/300 nouns from EK-100 also appear in ImageNet21k). As noted by Verbs-In-Action paper [3], verbs are relatively scarce in existing datasets. This explains why methods [4, 5] that use large-scale datasets excel at noun classification. On the other hand, our approach doesn’t use large-scale pre-training datasets and learns long-range spatio-temporal dynamics to achieve absolute SOTA on verb classification. Hence, we argue that our contribution is orthogonal to existing works per Reviewer UhsZ, and we make progress on a different aspect of the problem than data-scaling, namely long-context spatiotemporal actions.
>
>
> ### Q1. “Why does the approach only make sota for the verbs?”
> Please see above discussion, and also find related discussion in lines 297-304 in the main paper. Performing well on nouns requires understanding object semantics which are benefited by large-scale pre-training data, while performing well on verbs requires spatiotemporal understanding. The focus of our paper is on the second question.
>
>
> ### Q2. “... The approach needs to be tested on longer sequence videos”
> We extend the temporal context of MAE encoders from typical 16 frames to 128 frames. We found that reconstructing <15% of tokens can lead to performance drop, and with 15% we are already at the limit of the high-bandwidth memory (HBM) of the TPU accelerators at 128 frames. Orthogonal works use sliding window inference [6, 7], non-differentiable external memory [8, 9], frame filtering etc [10] to scale 16 frame video features to longer sequence tasks. In this paper, we demonstrate the utility of pretraining MAEs with longer context and provide novel techniques for doing so efficiently.  It is exciting future work to combine what we have discovered with these other orthogonal approaches to further scale up to even longer videos.
>
> ### Q3. “There is limited innovation on the actual network used”
> On the encoder part, it is our intended choice to use a well-known model architecture as vision transformers are the model of choice for video foundation models [11, 12, 13], allowing potential adoption of our method. While MAE architecture has minimal modification, our dual innovation is in (1) designing an adaptive tokenizer and (2) designing a methodology of using importance score from adaptive tokenizer for latent reconstruction to achieve the required decoder efficiency while maintaining / improving performance.
>
> We refer the reviewer to the qualitative results in Figure 3 and Figure 4 of our Appendix that shows that our method of selecting tokens is indeed superior to random selection or flow based methods, and this is corroborated by our empirical results from Table 1 of the main paper.
>
> ### Q4. “What is the limit on the length of tokens”
> We limit the number of decoder tokens to be in the same ballpark as the number of encoder tokens in our MAE pre-training setup, so that the memory, FLOPs and training time are not disproportionately affected by the decoder. Typically, MAE encoders use only 10% of the video tokens and are trained for 1600 to sometimes 3200 epochs [14]. However, as shown in Figure 1 (right) of our main paper, decoder memory and FLOPs disproportionately dominate at 128 frames making such training inefficient. Hence, it is crucial to limit the number of tokens to be approximately same as the number of encoder tokens during pre-training.
>
>
>
> ### Q5. “... dataset performance doesn't seem to make it worth it.”
> In Tables 3 (a) and 3 (b), we show SOTA performance on EK-100 verbs and Diving48. We further improved these numbers leading up to the rebuttal as shared in the common response. Moreover, we pre-train in an unsupervised manner without relying on large-scale datasets compared to related approaches.
>
>
> [1] Ridnik, Tal, et al. "Imagenet-21k pretraining for the masses." arXiv preprint arXiv:2104.10972 (2021). \
> [2] Grauman, Kristen, et al. "Ego4d: Around the world in 3,000 hours of egocentric video." In CVPR. 2022. \
> [3] Momeni et al., “Verbs in Action: Improving verb understanding in video-language models”. In ICCV 2023. \
> [4] Zhao, Yue, and Philipp Krähenbühl. "Training a large video model on a single machine in a day." arXiv preprint arXiv:2309.16669 (2023). \
> [5] Xiong, Xuehan, et al. "M&m mix: A multimodal multiview transformer ensemble." arXiv preprint arXiv:2206.09852 (2022). \
> [6] Wang, Xiang, et al. "Proposal relation network for temporal action detection." arXiv preprint arXiv:2106.11812 (2021). \
> [7] Chen, Guo, et al. "Video mamba suite: State space model as a versatile alternative for video understanding." arXiv preprint arXiv:2403.09626 (2024). \
> [8] Wu et al., “Memvit: Memory-augmented multiscale vision transformer for efficient long-term video recognition”. In CVPR 2022. \
> [9] Balazevic et al., “Memory Consolidation Enables Long-Context Video Understanding”. arXiv 2402.05861. \
> [10] Tan et al., “Koala: Key frame-conditioned long video-llm”. In CVPR 2024. \
> [11] Wang, Yi, et al. "Internvideo: General video foundation models via generative and discriminative learning." arXiv preprint arXiv:2212.03191 (2022). \
> [12] Li, Kunchang, et al. "Unmasked teacher: Towards training-efficient video foundation models." In ICCV 2023. \
> [13] Zhao, Long, et al. "Videoprism: A foundational visual encoder for video understanding." arXiv preprint arXiv:2402.13217 (2024). \
> [14] Tong, Zhan, et al. "Videomae: Masked autoencoders are data-efficient learners for self-supervised video pre-training." Advances in neural information processing systems 35 (2022): 10078-10093.

---

> ### Author Response · Authors · 2024-08-10
> **Please let us know whether we address all the issues**
>
> Dear reviewer,
>
> Thank you for the comments on our paper.
>
> We have submitted the response to your comments and a common response. Please let us know if you have additional questions so that we can address them during the discussion period. We hope that you can consider raising the score after we address all the issues.
>
> Thank you

---

> > ### Comment · Reviewer_9WAZ · 2024-08-12
> >
> > ok thanks for this detailled response, especially about the nouns/verb issue, I still have a concern about the length of the video sequences that are possible to be used though

---

> > > ### Author Response · Authors · 2024-08-13
> > > **Addressing the length of video sequences**
> > >
> > > Dear reviewer,
> > >
> > > Thanks for acknowledging our response, and we are encouraged that we have addressed your concerns on nouns/verbs. To demonstrate our model’s capability on longer video sequences, below, we provide a break-down of the performance improvement of our model over previous SoTA [1] on EPIC-Kitchens-Verbs based on video durations.
> > >
> > > |Model|0s-4s|4s-8s|8s-16s|16s-32s|>32s|
> > > |-------|------|-------|--------|---------|------|
> > > |AVION - Large (previous SoTA)|75.6|66.0|64.7|66.3|51.9|
> > > |Ours - Large|77.8|67.0|66.2|72.2|57.7|
> > > |Relative Improvement (%)|+2.9|+1.5|+2.3|+8.9|+11.2|
> > >
> > > We observe sustained performance improvements on videos with longer durations signifying our model’s capability on longer sequences. When we consider the number of frames, compared to contemporary video encoders [2, 3, 4], we can process 8X their number of frames. As stated in previous comments, we leave the exciting opportunity of using our video encoder as a backbone of an e2e long-video understanding system [5, 6] or adding a memory module to increase sequence length [7] further to the future works.
> > >
> > > ### More evidence on Nouns vs Verbs and further improvements on EK-Nouns
> > >
> > > On a side note, to augment our earlier point that EK-100-Noun can indeed be improved by large-scale pre-training, we post-trained our best model on labeled Kinetics710 dataset containing approximately 1M videos, and found that we can improve EK-100 Noun accuracy from 59.5 → 61.8% and retain our SoTA on EK-100 Verbs at 75.0%. Overall, our action classification accuracy is now 52.1% which places us at Rank 3, only lagging behind AVION [1] and M&M [8]. AVION uses 4M Ego4D clip-text pairs and M&M uses 60M clip-text pairs in their pretraining. Note that our Verb accuracy didn’t improve with large-scale pre-training datasets and we achieve SoTA on Verbs despite using an order of magnitude less data than these two methods. This matches our intuition that EK-100 Nouns and EK-100 Verbs require different expertise. We will include add this to our discussion in the final submission and we thank the reviewer for pointing the difference out.
> > >
> > > [1] Zhao, Yue, and Philipp Krähenbühl. "Training a large video model on a single machine in a day." arXiv preprint arXiv:2309.16669 (2023).
> > > [2] Wang, Limin, et al. "Videomae v2: Scaling video masked autoencoders with dual masking." Proceedings of the IEEE/CVF Conference on Computer Vision and Pattern Recognition. 2023.
> > > [3] Wang, Yi, et al. "Internvideo2: Scaling video foundation models for multimodal video understanding." arXiv preprint arXiv:2403.15377 (2024).
> > > [4] Ryali, Chaitanya, et al. "Hiera: A hierarchical vision transformer without the bells-and-whistles." International Conference on Machine Learning. PMLR, 2023.
> > > [5] Chen, Guo, et al. "Video mamba suite: State space model as a versatile alternative for video understanding." arXiv preprint arXiv:2403.09626 (2024).
> > > [6] Sun, Yuchong, et al. "Long-form video-language pre-training with multimodal temporal contrastive learning." Advances in neural information processing systems 35 (2022): 38032-38045.
> > > [7] Wu, Chao-Yuan, et al. "Memvit: Memory-augmented multiscale vision transformer for efficient long-term video recognition." Proceedings of the IEEE/CVF Conference on Computer Vision and Pattern Recognition. 2022.
> > > [8] Xiong, Xuehan, et al. "M&m mix: A multimodal multiview transformer ensemble." arXiv preprint arXiv:2206.09852 (2022).

---

> > > > ### Author Response · Authors · 2024-08-14
> > > >
> > > > We hope that we have addressed your concerns and would like to request you to reevaluate our work in light of our rebuttal.

---

### Official Review · Reviewer_LVuv · 2024-07-16

**Soundness:** 3
**Presentation:** 2
**Contribution:** 3
**Rating:** 6
**Confidence:** 4

**Summary:**

This paper focuses on efficiently extending VideoMAE to much longer videos. It proposes an adaptive decoder masking strategy that utilizes a  MAGVIT tokenizer to localize the importance of each token, which becomes targets that reduce the memory/computation of decoders.
The motivation aims to scale the input video along the temporal dimension while maintaining pre-training efficiency. The key innovation is using a tokenizer to generate masks and targets, significantly reducing the training burden. The evaluation mainly focuses on long-video recognition tasks like (Epic-Kitchens-100 and Diving-48).

**Strengths:**

The idea of using tokenization to select important tokens seems novel to me and can significantly improve the decoder's training efficiency in VideoMAE.
The ablation of Table 1/2/4 clearly shows each component's effectiveness, while there is space for improvement.
SoTA performance achieved without using a larger-scale dataset and under an efficient training budget setting seems promising.

**Weaknesses:**

My main concern is the fixed budget setting (128/32 frames and 15% masking ratio on two datasets), which limits the potential to become a standard baseline for further research. This method can work well with models of various sizes and longer videos, so it is important to show that it also performs well in standard VideoMAE settings (short-video datasets). As the model grows larger, the computation of the decoder increases not only with the number of frames but also with the model size. Therefore, I am eager to see the proposed method’s results (efficiency/performance) on different datasets, number of frames, and mask ratios. In short, I am not convinced by the current results that the proposed masking strategy can be a future research baseline.

The quality of the tokenizer is another important aspect that needs to be included. Table 4 shows that using MAGVIT as a target can significantly improve performance, indicating there are insights behind this choice. Moreover, the novel token selection module may influence the tokenization results or potentially benefit reconstruction. However, I cannot find tokenizer-related results. Specifically, why does utilizing such a tokenizer significantly reduce the number of tokens? Why choose MAGVIT instead of other options? The 3D tokenizer compresses the temporal dimension several times; could this lead to inferior performance?

**Questions:**

Please see above weakness.

**Limitations:**

Yes.

---

> ### Author Rebuttal · Authors · 2024-08-07
>
> We thank the reviewer for the detailed review and for recognizing the novelty, effectiveness and impact of our proposed approach. While we designed our experiments to study the impact of long-video MAE pre-training as such has not been attempted before, we agree with the reviewer that the proposed model can work well across model sizes and video lengths and we also agree that our justification on the choice of tokenizer is lacking. We hope that our below response addresses your concerns:
>
> ## Q1: Short-video datasets
> We favored datasets containing longer action sequences as they can benefit the most from the increased context-length. In addition, we present a study on Something-Something-V2 dataset in the [common response](https://openreview.net/forum?id=bFrNPlWchg&noteId=nljvnUQeki) where we show the same trend as observed in Table 2 of the main paper.
>
> ## Q2: Scale model size vs scale number of frames, given our decoder efficiency
> This is an interesting direction that we haven’t considered in our study as we focused on how to scale the number of frames. Below, we present some preliminary findings where we study the effect of varying model sizes vs varying the number of frames. For this experiment, we first fix the memory budget and masking ratio to the reference base model trained with 128 frames at 15% decoder masking, and try to maximize the number of frames that we can fit in this memory budget. We report final accuracy and FLOPs relative to the base model.
>
> |Model Size|GFLOPs (relative)|Number of frames with constant memory budget| EK-100 Noun-Verb accuracy|
> |----|----|---|----|
> |Small|0.43x|144|39.1|
> |Base (reference)|1x|128|47.3|
> |Large|1.52x|80|48.9|
>
> We find that with larger model size and fixed memory budget the accuracy improves despite lower frames but at the cost of significantly increased compute. We will include this discussion in our paper.  Finally, we’d like to reiterate that all the three settings above are only made possible due to the memory savings from our proposed strategy.
>
> ## Q3: Different number of frames and mask ratios
> Our motivation is to study whether we can effectively pre-train MAE on long-videos and whether such pre-training helps. For this, we selected 128 frames as our target (as it is 8x the typical MAE pre-training frames already), and designed our method around achieving this target. As shown in Figure 1 of the main paper, at 15% masking ratio, the decoder memory becomes less than encoder memory and hence we chose it as our masking ratio. Post-facto, we agree that it’s interesting to study the effect of varying number of frames and mask ratios. To satisfy reviewer’s curiosity, we present some related findings:
>
> ### Varying mask ratio
>
> We ablate the effect of the decoder mask ratio on accuracy while keeping the encoder mask ratio fixed at 10%. For this experiment we use our adaptive masking strategy, a 128 frame context for pre-training and fine-tuning and use adaptive MAGVIT tokens as pre-training targets. A 15% mask ratio yields the highest accuracy, significantly outperforming even the much more expensive setting with 50% mask ratio that requires >4x more memory than the 15% setting.
>
> |Decoder Masking|Diving-48 Accuracy|Memory usage (w.r.t. 15% masking ratio)|
> |----|----|----|
> |5%|88.1|N/A|
> |15%|89.7|1|
> |25%|88.7|1.25|
> |50%|87.5|4.2|
>
>
>
> ### Varying number of frames
> In the table below, we ablate the effect of increasing the number of frames while keeping the mask ratio fixed at 15%. During inference, we ensure that the model sees the same number of frames and report Epic-Kitchens-100 Top1 Noun-Verb Accuracy. We clearly observe a trend of the accuracy improvement with the number of frames similar to what we presented in Table 2 of the main paper.
> Note that both 64 and 128 frame MAE models go OOM without our decoder masking strategy.
>
> |Pre-training frames|Masking|Eval protocol|Epic-Kitchens-100|
> |-----|---|---|---|
> |16|15%|16 x 8 clips|41.7|
> |32|15%|32 x 4 clips|45.0|
> |64|15%|64 x 2 clips|47.1|
> |128|15%|128 x 1 clip|47.3|
>
>
>
> ## Q4: Tokenizer clarifications
> ``Why does utilizing such a tokenizer significantly reduce the number of tokens?``
>
> We believe that the tokenizer can be used to reduce tokens due to two reasons: (1) When we compare with pixels as targets, tokens can encode more surrounding context, and thereby they can have more redundancy. This choice allows us to afford higher masking ratios than used by VideoMAE-V2 [1] with pixels as targets (2) Our adaptive tokenizer training process forces the tokenizer to learn a good importance weighting scheme that gets reflected in the adaptive mask. This lets us have a high masking ratio. In addition, since our mask is learnt independently of MAE, unlike EVEREST [2], our masking scheme outperforms all the other schemes as shown in Table 1 of the main paper.
>
> ``Why choose MAGVIT instead of other options?``
> Please refer to the common response where we discussed the tokenizer and quantizer.
>
> ``The 3D tokenizer compresses the temporal dimension several times; could this lead to inferior performance?``
> Temporal downsampling is an inherent result of using the 3D-CNN MagViTv2 architecture (which reduces resolution but provides spatiotemporal features). We have not ablated the effect of temporal compression in the tokens on the downstream performance. However, we believe compression doesn’t necessarily lead to performance drop, as prior works [3] have shown that reconstructing high-level targets in MAE is better than reconstructing low-level targets and this is also corroborated by Table 1 of our main paper.
>
> [1] Wang, Limin, et al. "Videomae v2: Scaling video masked autoencoders with dual masking." Proceedings of the IEEE/CVF Conference on Computer Vision and Pattern Recognition. 2023.\
> [2] Hwang, Sunil, et al. "EVEREST: Efficient Masked Video Autoencoder by Removing Redundant Spatiotemporal Tokens."\
> [3] Yu et al. “Language Model Beats Diffusion -- Tokenizer is Key to Visual Generation”. In ICLR 2024.

---

> ### Author Response · Authors · 2024-08-10
> **Please let us know whether we address all the issues**
>
> Dear reviewer,
>
> Thank you for the comments on our paper.
>
> We have submitted the response to your comments and a common response. Please let us know if you have additional questions so that we can address them during the discussion period. We hope that you can consider raising the score after we address all the issues.
>
> Thank you

---

> > ### Comment · Reviewer_LVuv · 2024-08-12
> > **Response to rebuttal**
> >
> > Thanks for the authors' rebuttal. It addressed most of my concerns. The scaling of model size seems promising. And additional results on Something-something are helpful. I have one concern: What is the MagViTv2 inference cost regarding memory and speed? Will it become a bottleneck if we further extend the number of frames?

---

> > > ### Author Response · Authors · 2024-08-14
> > > **Thanks for your response**
> > >
> > > Dear reviewer
> > >
> > > Thank you for responding to our rebuttal and we are glad to have addressed most of your concerns. We will incorporate your several feedbacks in the final version! Below, we answer your follow-up questions:
> > >
> > > `What is the MagViTv2 inference cost regarding memory and speed?`
> > >
> > > We benchmarked MagViTv2 inference cost using a single TPUv4 device. We are able to use a maximum batch size of 32 with each batch element containing one clip of 16 frames. The peak memory usage is 6.4Gi and average throughput is 195 videos/sec. For longer video clips, as mentioned in lines 812-813 of the main paper, we simply slide a window of 16 frames over the entire video with a stride of 16 frames. So, the inference costs would scale linearly with video lengths while the memory requirement stays the same.
> > >
> > >
> > > `Will it become a bottleneck if we further extend the number of frames?`
> > >
> > > No, MagViTv2 will not become a bottleneck for longer videos because MAE training costs scale quadratically with video length while MagViTv2 costs scale linearly. In addition, MagViTv2 tokens can be computed and stored offline for the entire dataset just once as we do not update the tokenizer during MAE training. Note that we typically train MAEs for thousands of epochs, so the amortized cost of MagViTv2 computations is an order of magnitude less if we precompute the tokens.

---

### Author Rebuttal · Authors · 2024-08-07

## Summarizing feedback
We thank all the reviewers for their valuable time and feedback. We are encouraged by the positive feedback from all the reviewers who found our work
- Novel (Reviewer LVuv, Reviewer UhsZ), orthogonal to prior work (Reviewer UhsZ), and addresses an important issue in video understanding (Reviewer UhsZ),
- Solves the OOM problem of long-video training and beats previous masking strategies (Reviewer EcJb), saves computational overhead and improves model performance (Reviewer UhsZ), significantly improves the decoder’s training efficiency in VideoMAE (Reviewer LVuv)
- Achieves SoTA without using large-scale dataset (Reviewer LVuv), improved scores using a pretty standard network and pretraining (Reviewer 9WAZ), Compared to 32 frames, the proposed method with 128 frames obtains better performance (Reviewer EcJb)
- Verifies the effectiveness of each part through ablations (Reviewer UhsZ, Reviewer LVuv)
- Is well-written and easy to follow, with clear explanations of the proposed method and experimental results (Reviewer UhsZ)

## Further improvements to results:

Leading up to the rebuttals, we discovered that some of our hyper-parameter choices such as learning rate, whether to use cls token during fine-tuning etc. were not optimal, and we have obtained even better results than reported in the main paper (93.7 → 94.9 on Diving48 +3.9 over SoTA, and 75.0 → 75.5 on EpicKitchens-100 Verbs +2.5 over SoTA).


## Addressing common concerns:
We appreciate the feedback and the questions, and we address some common concerns below and address specific questions and concerns of the reviewers in the respective rebuttals. We hope that these adequately answer the reviewers’ concerns and we are happy to continue discussions, and delve into any details regarding our proposed method.

### Limited number of datasets and task types:

During the rebuttal period, we additionally tested our method on a short-clip dataset, Something-Something-v2, which has an average clip length of 3.8 seconds at 12fps, i.e. 46 frames using a Base sized model.

|Number of frames|Decoder Masking|Eval protocol|SSv2 val accuracy|
|-------|-----|-----|-----|
|16|None|16 x 4 clips|67.4|
|32|None|32 x 2 clips|69.9|
|64|15%|64 x 1 clips|71.0
|96|15%|96 x 1 clips|70.6

The results match the trends we have seen on the other datasets in Table 2 of the main paper, with longer context providing a significant boost in final accuracy. There are diminishing returns from using more than 64 frames of context, as only a small number of examples from this dataset have clip length this long.  Note that our performance exceeds both VideoMAE v1 [1] and VideoMAE v2 [2] for the base sized model, despite us pre-training for fewer epochs (1600 vs. 2400) and using only a single crop at evaluation time.

Due to time/resource constraints of the rebuttal period we were not able to run similar ablations for models with a VIT-L backbone and other downstream tasks, but hope to include these experiments as well in a future version. We will include this discussion in the final version of the paper.

### Choice of tokenizer and quantizer:

We chose MagViTv2[3, 4] (which has a 3D CNN based encoder) because it is a strong tokenizer architecture which has been shown to work both for generation and understanding [4] compared with prior works. On the choice of quantizer, prior works show that Lookup Free Quantizer (LFQ) [4] and Finite Scalar Quantizer (FSQ) [5] outperform traditional Vector Quantization (VQ) [6]. To study the effect of quantizer on our long-context MAE task, we performed several experiments comparing LFQ vs FSQ with MagViT as the tokenizer, and found that there is not much difference on the final EK-100 accuracy with these choices. However, FSQ with 18 bit codebook size showed the highest PSNR, the highest EK-100 accuracy and the second best FVD. So, we chose it as our quantizer. The corresponding study is presented below and we will include this in our final submission.

|Tokenizer|PSNR [K600]|FVD [K600]|Epic-Kitchens-100|
|---|---|---|----|
|LFQ 14 bit codebook|23.2|19.4|46.3|
|LFQ 18 bit codebook|22.6|14.7|46.2
|FSQ 14 bit codebook|24.3|24.3|45.1
|FSQ 16 bit codebook|24.7|21.9|46.0
|FSQ 18 bit codebook|25.1|19.1|46.4



[1] Tong, Zhan, et al. "Videomae: Masked autoencoders are data-efficient learners for self-supervised video pre-training." Advances in neural information processing systems 35 (2022): 10078-10093. \
[2] Wang, Limin, et al. "Videomae v2: Scaling video masked autoencoders with dual masking." Proceedings of the IEEE/CVF Conference on Computer Vision and Pattern Recognition. 2023. \
[3] Yu et al. “Magvit: Masked generative video transformer.” In CVPR 2023. \
[4] Yu et al. “Language Model Beats Diffusion -- Tokenizer is Key to Visual Generation”. In ICLR 2024. \
[5] Mentzer, Fabian, et al. "Finite scalar quantization: Vq-vae made simple." arXiv preprint arXiv:2309.15505 (2023). \
[6] Yu, Jiahui, et al. "Vector-quantized image modeling with improved vqgan." arXiv preprint arXiv:2110.04627 (2021).

---

### Decision · Program_Chairs · 2024-09-25

**Decision:**

Accept (poster)

**Comment:**

The reviewers praised the paper's novelty, writing style, and clear explanations of the proposed method and experimental results. They also acknowledged the importance of addressing the scalability of video masked modeling for longer sequences.

While some reviewers initially expressed concerns about the limited scope of the experiments, the authors have since provided additional results on an additional dataset (SSv2), number of frames, and mask ratios, demonstrating the robustness of their approach. The authors have also clarified the quality of the tokenizer used and its impact on performance. Furthermore, the authors have addressed the concern about testing on longer video sequences, providing new results that show the effectiveness of their approach on longer sequences.

Overall, the authors have successfully addressed the concerns raised by the reviewers, providing additional experiments and explanations that strengthen the paper's contributions. The paper has undergone significant improvements since the initial submission, and the authors have demonstrated a thorough understanding of the reviewers' concerns.

After carefully reviewing the reviews and rebuttal, this meta-reviewer concluded that this paper is ready for publication and will make a valuable addition to the research community.